# Effectiveness of Standardized Protocol for Oxygen Therapy on Improving Nurses’ Performance and Patients’ Health Outcome

**DOI:** 10.3390/ijerph19105817

**Published:** 2022-05-10

**Authors:** Samar Salah Eldin Mohamed Diab, Shaimaa Ahmed Awad Ali, Shaymaa Najm Abed, Gehan Abd Elfattah Atia Elasrag, Osama Mohamed Elsayed Ramadan

**Affiliations:** 1Department of Nursing, College of Applied Medical Science, Jouf University, Sakaka 2014, Saudi Arabia; saaali@ju.edu.sa (S.A.A.A.); snabed@ju.edu.sa (S.N.A.); gehanatia@ju.edu.sa (G.A.E.A.E.); omramadan@ju.edu.sa (O.M.E.R.); 2Pediatric Nursing Department, Faculty of Nursing, Menoufia University, Shebin El-Kom 32511, Egypt; 3Critical Care and Emergency, Faculty of Nursing, Mansoura University, Mansoura 35516, Egypt; 4Adult Health Nursing (Medical-Surgical Nursing), Menoufia University, Shebin El-Kom 32511, Egypt; 5Pediatric Nursing Department, Faculty of Nursing, Cairo University, Cairo 11562, Egypt

**Keywords:** standardized protocol, oxygen therapy, nurses’ performance, patients’ health outcome

## Abstract

Aims: assess nurses’ knowledge and performance-related safe administration of oxygen (O2) therapy; apply an intervention program for nurses about standardized protocol for oxygen; and evaluate the effectiveness of standardized protocol for oxygen in improving nurses’ performance and patients’ health outcomes. Design: a quasi-experimental study was used. Setting: the current study was conducted at three hospitals in Sakaka City with totally different medical aid units (ICUs), CCUs, emergency care departments (ED), medical and surgical wards, pediatric care units (PICUs), neonatal intensive care units (NICUs), pediatric emergency care departments (PED) and pediatric inpatient\outpatient departments. Subjects: a convenience sample of 105 nurses and 105 patients was divided into 55 patients in the control group who received routine care and 50 patients in the study group who received intervention. Findings: 34.3% of studied nurses had poor knowledge pre-intervention compared with 17% post-intervention. Moreover, 33.3% of them had satisfactory knowledge pre-intervention versus 21% post-intervention. Only 5.7% of them had excellent knowledge pre-intervention, compared with 34.4% post-intervention. Concerning the complications of oxygen therapy, only 10.5% did not have complications in the control group versus 62.9% in the study group, 33.3% of the control group had cyanotic lips and fingernails pre-intervention, versus 7.6% in the study group; 10.5% had oxygen toxicity in the control group, versus 7.6% in the study group, with a highly statistically significant difference at p 0.001 for all. Conclusion: the current results of this study concluded that there was improvement in nurses’ knowledge and practice related to oxygen therapy post-intervention. Moreover, when the standard protocol for safe oxygen therapy was used in a positive way, it led to better health for patients and fewer problems with oxygen therapy.

## 1. Introduction

Supplemental oxygen therapy (SOT) is a medicinal treatment for tissue hypoxia. When used correctly, it has the ability to improve medical results and save lives, but when used incorrectly, it has the potential to damage people. A fundamental item in the World Health Organization’s (WHO, Geneva, Switzerland) model of essential drugs used in a healthcare system is oxygen. SOT is crucial for avoiding and controlling hypoxemia in both acute and chronic circumstances. SOT is considered a key tool for resuscitating patients during a general assessment. According to WHO updated guidelines, if given at the correct time and in the proper amount, SOT can help save the lives of many people with heart and lung diseases [1].

Oxygen is one of the first all-encompassing medications for people suffering from a variety of ailments. In most emergency circumstances, oxygen is employed. Medical help with little oxygen can be lethal. As a result, patients must be treated safely and comfort ably [2,3]. To keep important organs receiving adequate oxygen, more oxygen, often in high amounts, is required. Although oxygen therapy can save lives, it can also be detrimental if used in high doses for an extended period of time. The oxygen saturation level for many acutely sick patients should be 94–98%, or 88–92% for those at risk of hypercapnic metabolic failure [4].

Higher oxygen concentrations, like any medicine, can injure the human body and cause major health problems. The most prevalent adverse impact of high oxygen concentrations is oxygen poisoning [5,6]. Any errors in oxygen medical assistance will exacerbate a patient’s condition and be fatal. Oxygen medical assistance is an important part of resuscitation, acute medical treatment, life-supporting oxygen, and anesthetics. Although the hazards and benefits of oxygen therapy are well-known, oxygen medical care is frequently administered by members of the health care team who lack specialized training, adequate information, or ample practice opportunities [7,8]. 

Health team members are performing a crucial role while administering oxygen (O2), which should be done by physicians or nurses. Nurses play a vital role in patient monitoring and need to be familiar with O2 therapy safe administration side effects and intervention for complication. Prescribed initial investigations like ABG and chest X-ray should be done frequently and assessed rigorously as the observation of significant signs, level of consciousness, and pulse-oximeter are required. An alert nurse should be aware of the physician’s prescription for oxygen therapy, including indications, target oxygen saturation, oxygen delivery device, and range of (O2) flow or share (galvanized O2), and when it is to be administered. The prescription should be signed and dated by the doctor [4].

Because oxygen medical aid is administered to patients in the same manner as any other drug, nurses may experience some challenges while administering it. It is likely that these roadblocks can be linked back to the nurses who use the technology, such as a lack of understanding of the various aspects. There may also be obstacles connected to the hospital, such as a shortage or unavailability of appropriate working devices or supplies used during element medical assistance, or a lack of protocol for element medical assistance. All of these are examples of impediments. There are also some other obstacles that could be related to prescription, such as a physician’s prescription that is unclear on dosage, or a gadget that should be used according to the instructions.

Oxygen therapy is a crucial medical treatment for people of all ages and conditions who require it. Patients are at risk for a range of major health concerns, including hypoxemia, respiratory pathology, and even death, if oxygen medical treatment is not delivered properly [9]. As a result, it is vital for caregivers to guarantee that chemical element medical help is administered precisely, safely, and on time. As a result, the goal of this study was to see whethera consistent strategy for oxygen medical assistance may improve nurses’ performance and patients’ health outcomes.

### 1.1. Research Hypotheses

There are positive effects of a standardized protocol for oxygen therapy utilization and improving nurses’ knowledge and performance.Competent nurses’ administration of standardized care for oxygen therapy will improve the patients’ outcomes in the study group.

The current study was conducted at three hospitals in Sakaka town in different medical aid units (ICUs), CCU, emergency care departments (ED), medical and surgical wards, pediatric care units (PICUs), neonatal intensive care units (NICUs), pediatric emergency care departments (PED) as well as pediatric inpatient and outpatient departments.

A convenience sample of 105 nurses and 105 patients was determined using a random assignment method. Patients were divided into 55 patients in the control group who received routine care and 50 patients in the study group who received interventions. The nurses who partook in this study operated within the three hospitals in Sakaka city and consented to participating in the study. They worked in different medical aid units (ICUs), CCU, emergency care departments (ED), medical and surgical wards, pediatric care units (PICUs), neonatal intensive care units (NICUs), pediatric emergency care departments (PED), as well as pediatric inpatient and outpatient departments.

Sample Size: the sample size was calculated based on the statistical power of 90% and level of confidence (1-Alpha Error): 95%, Alpha 0.05, Beta 0.1. Every group determined the sample size, which was set at 30 patients. Considering 15% sample attrition, the final sample size in the study group was 50 and the control group 55.

### 1.2. Data Assessment Tools

Part I: Characteristics of the studied nurses such as age, gender, qualifications, years of experience, workplace, attended training courses, and type of shift [10].Part II: Knowledge level: This was developed by researchers after reviewing literature reviews as Jacobs et al [11]. This part included 18 close-ended questions in a MCQ form divided into five domains: concept of oxygen (4 questions), principle and indications of oxygen therapy (4 questions), oxygen toxicity (3 questions), nurse role-related prevention of oxygen toxicity (3 questions) and nurse role-related oxygen therapy (4 questions). Nurses’ responses scored as one point for a correct answer and zero for an incorrect answer; the level of knowledge is considered good if nurses score >75%, average if the score is between 60 and 75%, and poor if the score <60%.Part III: Observation performance level: this was developed by researchers after reviewing literature reviews by researchers such as Ford & Robertson [12]. This part included three stages: preparation of oxygen therapy, provision of oxygen therapy, and reporting. If done, the item scored as one point, and zero if not done.Part IV: Assess patients’ outcome: This was developed by researchers after reviewing literature reviews by researchers such as Nguyen et al., Mostfa et al, Hemati et al and Hendy et al. [1,13,14,15]. It included duration of hospital stay, duration of oxygen therapy and complications.

Pilot study: A pilot study was conducted on a group of 11 nurses (10%). It was conducted prior to data collection to assess the feasibility and duration of data collection. No modification was carried out, therefore, the participants in the pilot were included in the study.

Validity: A group of five experts in critical and medical/surgical nursing ascertained the content’s validity; their opinions were elicited regarding the format, layout, consistency, accuracy, and relevancy of the tools.

Reliability: The adapted tools were tested for their reliability by using Cronbach’s alpha coefficient test in SPSS program version 24 by a statistician. The Internal consistency reliability (Cronbach’s α) for part II was good (0.823), part III was good (0.839) and part IV was good (0.845).

Ethical Consideration: permission from the Bioethical committee was obtained, as was the permission of each dean of the faculty of applied life science and the head of the nursing department. The necessary official approvals were obtained from the directors of the available hospitals in Sakaka city to conduct the research according to the ethical guidelines of the National Ethics Committee, as well as the approvals of the nurses participating in the study.

### 1.3. Study Framework

This study was carried out during a period of six months from the beginning of June 2021 to January 2022. The researchers visited the hospital three times per week and collected the data during interviews with the nurses. The time needed to fill in the questionnaire sheet for every nurse was 30 min. Researchers asked the nurses to complete pretest questionnaires, observe their performance during pre-assessment, and assess outcomes among patients enrolled in the control group. Researchers divided nurses into five groups; each group trained for four sessions, each one lasting 45 min. Researchers informed nurses about their group and time of sessions, and sessions were conducted in the conference room of the hospital in coordination with the medical director of the hospital and nurse manager. Session content and education program was based on the reviewing of literature reviews. Researchers used different ways and illustrative methods such as PowerPoint, colored photo, animation, and short videos.

During the first session, the researcher explained the aim, significance and tool of the study, then, various topics including definition, different ways, indication and physiology, and pathology of oxygen therapy.

The second session included main topics such as concepts of oxygen toxicity, causes, signs of oxygen toxicity, management of oxygen toxicity, and the effect of applying standard oxygen protocol on patients’ outcomes.

During the third session, the researcher focused on the role of the nurse in providing oxygen therapy, the nurse’s role in preventing oxygen toxicity, and standardized protocol for safe oxygen therapy administration.

During the last session, the researcher summarized the training program, asked the nurses whether they had any questions and for feedback, and had an open discussion. Then, nurses were asked to complete a post-test questionnaire, the same one used pre-intervention, observe their performance, and assess the effect of the intervention on the patients enrolled in the intervention group. Researchers depended on multiple education methods such as group discussion, brainstorming and reflective thinking, and different illustrative methods such as PowerPoint, photos, and videos.

Data collected from the studied sample was revised, coded, and entered using a personal computer (PC). Computerized data entry and statistical analysis were performed using IBM SPSS Statistics for Windows, version 24 (IBM Corp., Armonk, NY, USA). Data were presented using descriptive statistics in the form of frequencies, percentages, and mean SD. A correlation coefficient “Pearson correlation” is a numerical measure of some type of correlation, meaning a statistical relationship between two variables. The Kruskal–Wallis H test (sometimes also called the “one-way ANOVA on ranks”) is a rank-based nonparametric test that can be used to determine if there are statistically significant differences between two or more groups of an independent variable on a continuous or ordinal dependent variable. The Wilcoxon test compares two paired groups and comes in two versions, the rank sum test, and signed rank test.

## 2. Results

Out of 105 nurses who responded to the questionnaire, 83 (79%) of them were in the range of 20–25 years of age. Among the participants, 94 (89.5%) were females. Moreover, 89 (84.8%) had a bachelor’s degree. Of all respondents, 56 (53.3%) had between 1–<3 years of experience. Moreover, 25 (22.3%) were specialized in emergency and adult ICU, 75 (71.4%) were single, and 85 (81%) had a morning shift (Table 1).

Table 2 revealed that the mean age of studied patients was 28.11 ± 5.57, whereas that of the control group was 28.67 ± 6.99; 69.1% of patients in the study group were male and68% in the control group. According to medical diagnosis, 63.8% of patients in the study group suffered from respiratory distress, compared with 66% in the study group.

Regarding the effectiveness of the standardized protocol on patients’ health outcomes, the mean hospitalization period was 12 ± 9.8 days within the range of 1–60-day for the control group, compared with 8.4 ± 6.2 within the range of 1–29-day for the study group with a highly statistically difference at *p* = 0.001. Additionally, the mean duration of oxygen therapy was 10.4 ± 8.9 days within the range of 1–36 days for the control group, compared to 7.2 ± 5.4 within the range of 1–28 days for the study group, with a highly statistically difference at *p* = 0.001. Concerning complications of oxygen therapy, in the control group, only 10.5% did not have a complication, versus 62% in the study group; 32.7% of the control group had cyanotic lips and fingernails versus 8% in the study group, with a highly statistically significant difference at *p* = 0.001 for all (Table 3).

Table 4 demonstrated that 34.3% of studied nurses had poor knowledge pre-intervention, compared with 17% post-intervention. Moreover, 33.3% of them had satisfactory knowledge pre- versus 21% post-intervention. On the other hand, only 5.7% of them had excellent knowledge pre-intervention, compared with 34.4% post-intervention. Regarding the performance score of the studied nurses, the mean of the grand total performance score was 49.6 ± 21.3 pre-intervention, versus 62.7 ± 10.1 post-intervention, with a highly statistically significant difference at *p* = 0.000. Regarding preparation, the mean performance score was 11.6 ± 5, compared with13.4 ± 3.2 pre/post-intervention. Regarding intervention, the mean performance score was 21.5 ± 9.3 versus 27.5 ± 4.6 pre/post-intervention. Moreover, regarding reporting, the mean performance score was16.6 ± 7.8, compared with 21.8 ± 2.9 pre/post-intervention. Markedly, there was a highly statistically significant difference at *p* = 0.000 for all.

Table 5 proved that there was a highly statistically positive linear correlation between the total scores for preparation, intervention, and reporting, and the total knowledge score at *p* value 0.001, 0.003, and 0.009, respectively. Additionally, there was a highly statistically positive linear correlation between the total performance score and total knowledge score at *p* value 0.000.

## 3. Discussion

Oxygen therapy is the administration of oxygen as a medical intervention, which can be for a variety of medical and surgical conditions. Patients can be affected by getting no oxygen, too little, or too much oxygen. It is necessary to provide the optimal concentration of supplemental oxygen to the acutely ill hypoxemic patient; inadequate oxygen administration may result in cardiac arrhythmias, tissue injury, renal injury, and ultimately, cerebral damage.

One of the most widely utilized healthcare remedies in the world is oxygen. Its widespread availability could imply that healthcare professionals (HCPs) are well-versed in its applications and limits. Oxygen is a medication, and nurses should be acquainted with the side effects, possible dangers, and proper oxygen therapy delivery [1,16]. The researchers stated that nurses need more training in oxygen treatment, and that suitable guidelines to guide oxygen therapy should be developed. Therefore, this study aimed to evaluate the effectiveness of standardized protocols for oxygen medical aid to improve nurses’ performance and patients’ health outcomes.

The importance of this study comes in the view of the wide and emergency use of O2. When utilized correctly, oxygen has the potential to save lives. Any mistakes in oxygen therapy, on the other hand, might exacerbate a patient’s illness and possibly be life-threatening [8,17,18].

Regarding the socio-demographic profile of the studied nurses, among a total of 105 nurses who responded to the questionnaire, more than two-thirds were in the range of 20–25 years old, and most of them were females with a bachelor’s degree. Of all respondents, slightly more than half had 1–< 3 years of experience. These findings agree with a quasi-experimental research study carried by Mostafa et al, O’Driscoll et al, and Katsenos, & Constantopoulos [13,19,20], who found that most of the nurses were females and had Bachelor of Nursing degrees. However, our results disagree with the previously mentioned study regarding years of experience, as the number of years of experience was between 5 and 10. These results may be explained by the fact that most of the participants were female, due to the recent introduction of males studying nursing. Because of the emergency and urgent nature of the departments where they are recruited, the majority of them have a bachelor’s degrees in nursing.

On the other hand, in their cross-sectional study, Aloushan et al, as well as Kelly and Michelle [21,22] reported that the male-to-female ratio was almost equal, with slightly more males. The majority of participants had 2–9 years of work experience.

The current study highlighted that nearly two-thirds of participants did not attend training or even related workshops; only 38% of participants attended such types of training or workshops. These findings were in line with ELgneid et al. [23] who revealed that more than one third (42.9%) attended oxygen therapy-related workshops, and of them, one quarter (25.7%) attended only one workshop. Additionally, other similar findings reported by Thabet et al. (2020) [24] showed that the majority of the studied nurses (80.0%) did not attend any previous training courses.

Regarding the effectiveness of the standardized protocol on patients’ health outcomes, the current study revealed a significant reduction in the hospitalization period and the duration of oxygen therapy usage in the study group compared to the control group. Moreover, there was also a more significant reduction in the complications of the oxygen therapy in the study group than the control group. These findings prove the effectiveness of the standardized protocol for oxygen therapy on improving the patients’ outcomes.

A similar finding reported by Wang et al. [25] revealed a better quality of life among COP patients receiving self-management education (SME). Additionally, Wang et al [25] reported a significant reduction in COPD-related hospital admissions and emergency department visits in the intervention group. Moreover, SME positively affects the reduction in COP patients’ emotional distress.

Regarding nurses’ knowledge, the current study showed that there were highly significant differences in knowledge mean scores between pre- and post-implementation of a standardized protocol for oxygen therapy (*p* = 0.000). This result is in line with Mostafa et al, as well as The Padma and Lakshmi [13,26], they illustrated a statistically significant difference in nurses’ knowledge pre- and post-implementation of an educational program about oxygen therapy at (*p* < 0.001). The total nurses’ knowledge mean score improved from (9.080 ± 4.818) before the educational program to (19.840 ± 0.421) after its implementation.

The low level of knowledge before the implementation of the study intervention may be related to the lack of the periodical training programs on oxygen therapy. This view is supported by Aloushan et al. [21] who declared that it is imperative that nurses be better informed on the dangers of giving oxygen therapy to their patients. Patients’ conditions and outcomes may be harmed if oxygen treatment is not used properly in certain life-threatening circumstances.

Furthermore, Markocic et al. [27] mentioned that in order to enhance their skills and avoid anticipated problems, nurses must be taught about oxygen management and particular requirements (oxygen toxicity). In addition, Aloushan et al. [21] concluded that there is a knowledge, attitude, and practice gap among health care workers when it comes to providing oxygen treatment to a patient, which may have an impact on the well-being of patients. It was also suggested that healthcare professionals require more exposure to oxygen treatment via comprehensive education and training programs.

Other coherent findings were demonstrated in a study carried by Mayhob [28]; more than two-thirds of the participants in the study lacked adequate understanding on how to give oxygen treatment. Only 6% and 18% of the participants in the study had good and average understanding on how to administer oxygen treatment, respectively.

A contradictory finding reported in a study carried by Kane et al. [7] concluded that there is a dearth of knowledge among COPD patients and the general public about the benefits and dangers of using oxygen. Health-care professionals are more knowledgeable with regards to oxygen therapy, but they are worried about a lack of training and equipment. 

Regarding nurses’ practices, the current study showed that more than one-third of studied nurses had risky performance pre-intervention, compared with 7.6% post-intervention. Additionally, only 8.6% had a good performance score pre-intervention, compared with 22.9% post-intervention. However, about half of them had excellent performance scores pre-intervention, compared with about two-thirds post-intervention. These findings reflect the effectiveness of the standardized protocol for oxygen therapy administration on improving nurses’ performance, which touches the core of the study about the safe administration of oxygen.

These finding are in line with Mostafa et al. [13], who illustrated that in total, approximately 14% of nurses had poor practice in the pre-test before the educational program was implemented, and 100% of nurses had acceptable practice after the educational program was implemented. A very statistically significant difference at *p =* 0.00 was found between the mean scores of all nurses before and after the implementation of the educational program, with the improvement being (63.04 ± 7.94101) to (97.92 ± 39.590). According to these results, most nurses had little opportunity to update their practice after they were established in the clinical setting, particularly in departments with high workloads.

Moreover, Mayhob [28] indicates that only 18% of the people in the study were using oxygen treatment at an acceptable level. Meanwhile, 40% of them had mediocre habits and 42% had insufficient ones. Adipa et al. [2] clarified that themes generated on commencement and oxygen treatment monitoring includes initiating oxygen therapy, as well as assessing and tracking the patient’s progress. The knowledge and information gap, lack of procedure, availability, and cost of delivery devices, as well as the oxygen supply were all issues related to oxygen difficulties.

Another supported view reported by Bunkenborg and Bundgaard [29] concluded that additional oxygen therapy for intensive care patients is handled by nurses based on daily physician-prescribed upper and lower pO2 and pCO2 limits, on nurses’ understanding of the individual clinical patient situation, and on knowledge of the benefits and drawbacks of oxygen therapy, including observational and clinical assessment expertise.

The current study revealed that there was a highly statistically positive linear correlation between the total performance score and total knowledge score at *p* value 0.000. This result was in harmony with Mostafa et al, Doyle and McCutcheon [13,30], who illustrated that there was a highly statistically positive correlation between total nurse’ knowledge and total practice after the implementation of educational programs (*p* < 0.001).

To sum up, the standardized protocol about safe oxygen therapy administration is effective in improving nurses’ knowledge and practices regarding O2 therapy and enhancing patients’ outcomes.

## 4. Conclusions

Finally, the current results of this study concluded that there was improvement in nurses’ knowledge and practice related to oxygen therapy post-intervention. Moreover, the positive application of the standardized protocol about safe oxygen therapy led to an improvement in patients’ conditions and a reduction in the complications of oxygen therapy.

### 4.1. Recommendations

The current study emphasizes on the safe application of therapeutic oxygen provided by nurses, as well as encourages the hospital authority to arrange continuous training courses and workshops to maintain the safe administration of oxygen therapy and standard quality of medical aid for patients receiving oxygen therapy. Moreover, a standardized protocol of oxygen medical aid should be utilized in hospital departments.

### 4.2. Limitation of the Study

The current study was conducted on sample size in Sakaka city solely, and it should be applied to all Jouf region to generalize the results for different hospitals. A convenience sample of 105 nurses and 105 patients was used, with patients divided into 55 in the control group who received routine care and 50 in the study group who received interventions. Due to the conditions of the Corona virus pandemic, the paper questionnaires as well as the observation forms were replaced with electronic ones with links provided for them to be filled out remotely. To give oxygen on a large scale, researchers were assisted by the heads of nursing in the units and departments selected to conduct the research during three shifts. Thanks and gratitude from the researchers to all the nurses participating in the study, as well as the heads of departments and units in the hospitals under study.

## Figures and Tables

**Table 1 ijerph-19-05817-t001:** Distribution of studied subjects in relation to socio-demographic and work characteristics (N = 105).

Nurses’ Socio-Demographic Characteristics	No	%
Age (years):		
20–25 Y	83	79
26–49 Y	22	11
Gender:		
Female	94	89.5
Male	11	10.5
Level of Nurse Education		
Bachelor	89	84.8
Nurse Diploma	14	13.3
Postgraduate	2	1.9
Experience in ICU		
<1 year	32	30.5
1–<5 years	56	53.3
5–10 years	15	23.4
Specialty:		
Emergency department	25	23.8
Adult ICU	25	23.8
Obstetric emergency	10	9.5
Pediatric (PICU and NICU)	17	16.2
Pediatric inpatients	24	22.9
Surgery Unite	4	3.8
Social status:		
Single	75	71.4
Married	23	21.9
Divorced/Widowed	7	6.7
Shifting:		
Morning	85	81
Afternoon	15	14.3
Night	5	4.7

**Table 2 ijerph-19-05817-t002:** Distribution of studied patients in relation to their characteristics (N = 105).

	Control		Study	
	N = 55		N = 50	
Socio Demographic Characteristics of studied patients	No	%	No	%
Age (Years):				
Infants (<1 year)	5	9.1	4	8
10–20	11	20	10	20
21–30	17	30.9	15	30
31–40	8	14.5	7	14
41–50	7	12.7	7	14
>50 years	7	12.7	7	14
Mean ± SD	28.67 ± 6.99		28.11 ± 5.57	
Gender:				
Male	38	69.1	34	68
Female	17	30.9	16	32
Medical diagnosis				
Respiratory distress	35	63.8	33	66
Use in resuscitation	7	12.7	6	12
Infection with fungi	13	23.6	11	22

**Table 3 ijerph-19-05817-t003:** The effectiveness of standardized protocol for oxygen medical aid on patients’ health outcomes.

Items	Control	Study	*p* Value
Duration of hospital stay/days			
X ± SD Range	12 ± 9.8 days 1–60 days	8.4 ± 6.2 1–29	W(_paired)_ = 5.3, *p* = 0.001 **
Duration of oxygen therapy/days			
X ± SD Range	10.4 ± 8.9 days 1–36 days	7.2 ± 5.4 1–28 days	W_(paired)_ = 3.2, *p =* 0.001 **
Patients’ complications relating to oxygen administration:			
	No. %	No. %	
No complications	6 10.9	31 62	K = 72.3, *p* = 0.000 **
Cyanotic lips and fingernails	18 32.7	4 8	
Slow, shallow, difficult, or irregularity breathing	14 25.4	10 20	
Infection with fungi	8 14.5	2 4	
O2 toxicity	9 16.4	3 6	

** high significant <0.01. W_(paired)_ = Wilcoxon test for non-parametric data for two related groups (same patients); K = Kruskal–Wallis test for non-parametric data for more than two related categories.

**Table 4 ijerph-19-05817-t004:** The effectiveness of standardized protocol for oxygen medical aid on participating nurses’ knowledge & performance pre- and post-intervention.

Knowledge Aspects	Knowledge Levels							
	Poor No (%)		Satisfactory No (%)		Good No (%)		Excellent No (%)	
Pre-intervention	36	34.3	35	33.3	28	26.7	6	5.7
Post-intervention	18	17	22	21	19	27.6	36	34.3
*p value*	*K = 22.7, p = 0.000 HS*							
Intervention phases	Pre		Post			*T* test		*p* value
	Mean ± SD		Mean ± SD					
Grand total	49.6 ± 21.3		62.7 ± 10.1			19.689		0.000 **
Preparation	11.6 ± 5		13.4 ± 3.2			21.644		0.000 **
Intervention	21.5 ± 9.3		27.5 ± 4.6			18.055		0.000 **
Reporting	16.6 ± 7.8		21.8 ± 2.9			20.614		0.000 **

** high significant <0.01. W_(paired)_ = Wilcoxon test for non-parametric data for two related groups (same nurses), K = Kruskal–Wallis test for non-parametric data for more than two related categories.

**Table 5 ijerph-19-05817-t005:** Correlation between studied variables (N = 105).

Items	Total Score of Preparation		Total Score of Intervention		Total Score of Reporting		Total Score of Performance	
	r	*p*	r	*p*	r	*p*	r	*p*
Total Score of Knowledge	0.642	0.001 **	0.577	0.003 **	0.497	0.009 **	0.711	0.000 **

** high significance if *p* value < 0.01.

## Data Availability

Not applicable.

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
