# Peer review of "Effectiveness of Standardized Protocol for Oxygen Therapy on Improving Nurses’ Performance and Patients’ Health Outcome"

_ijerph, 2022, doi:10.3390/ijerph19105817_

Round 1

Reviewer 1 Report

 This study could be of interest and of clinical relevance for the readers of  IJERPH. The paper is clearly written.The sample of pilot sudy is quite small. Please better explain the sample size calculation. Then, it would have been interesting to have more information on clinical characteristics and on some laboratory findings. 

Author Response

Sample Size: The sample size was calculated based on statistical power of 90%, level of confidence (1-Alpha Error): 95%, Alpha 0.05, Beta 0.1. Every group determines the sample size, which is set at 30 patients. Considering 15% sample attrition, the final sample size in study group is 50 and control group 55 patients.

Reviewer 2 Report

Dear Authors, it is very important paper, but the authors can describe the role of comorbidity in enrolled population.

1. The role of comorbidity is very important for the improvement of health status of enrolled population: For example, nurses or population with hypertension or diabetes (if there are present) have different responses to therapy?
2. The improvement described, is related to improvement in the health status, such as cefalea (stress symptoms) as described by Ciarambino et al 2021.
3. Is this population enrolled in 24-h shifts (8-14, 14-20, 20-08)?
4. This population is represented to a greater extent by female subjects. Is it a gender gap?

Author Response

Dear Reviewer 2 Thank you for your  valuable  comments 

Find attachment reply on comments 

Reviewer 3 Report

Abstract

The aim of the study in the abstract is not congruent with what was written in the introduction (the purpose of the study should be consistent everywhere)

The abstract needs to be rephrased to include more information about the assignment process of participants into control and intervention group (it is not clear) (did you use random assignment process) ---- this issue should also be illustrated in the method section ??

The abstract needs more improvements in relation to language structure. Some statement makes you confused, particularly in the results part.

Some used terms are not clear (e.g ..positive application ??,

Need to rephrase the following statement (long and confusing )

“Concerning complications of oxygen therapy, only 10.5% didn’t have complication control group versus 62.9% at study group, 33.3% of control group had cyanotic lips and fingernails pre intervention, versus 7.6% at study group, 10.5% had oxygen toxicity at control group, versus 7.6% at study group with highly statistically significant difference at p=0.001 for all. Conclusion”

Introduction

The significance of the study is not clearly illustrated

Line 86, the purpose of the study is written in future tense ???

Material and method

The purpose of the study is also repeated in the method part? Why?

Inconsistent purposes are placed in different parts of the study……. (is “Applying an intervention program “ is the purpose of the study? 

Different terms were used for the same concepts “Applying an intervention program” or “ Standarized protocol” , Can you differentiate between them ??

There is no need to mention questions and hypotheses…. (you either use questions or hypothesis ??)

Power analysis is highly needed in such studies to ensure that we have reached the required power of the test

Is there random assignment into groups ??

Line 113, spelling problems “Data assortment Tools:” (it should be assessment )

No information was provided about the reliability of the used scales (the mentioned Cronbach alpha is not accurately describe which subscale it reflects)

The process of recruiting nurses and patients is not clear ?? (it is a little confusing )

Need to illustrate how did the researcher sample the patients (the sampling method was only describing nurses)

Inclusion and exclusion criteria for nurses and patients are not illustrated

Results

The distribution of nurses into control and experimental groups is not illustrated ?

The researcher should explain in the method section the statistical test that has been used according to the level of measurement

Author Response

Point 1: The aim of the study in the abstract is not congruent with what was written in the introduction (the purpose of the study should be consistent everywhere

Evaluate the effectiveness of standardized protocol for oxygen medical aid to improve nurses' performance and patients' health outcomes.

Point 2: The abstract needs to be rephrased to include more information about the assignment process of participants into control and intervention group (it is not clear) (did you use random assignment process) ---- this issue should also be illustrated in the method section ??

No, the subjects enrolled at control and study group by non random way

Point 3: The abstract needs more improvements in relation to language structure. Some statement makes you confused, particularly in the results part.

To evaluate the effectiveness of standardized protocol for oxygen on improving nurses' performance and patients' health outcomes. Design: A quasi-experimental study was used.  Setting: The current study was conducted at three hospitals in Sakaka City with totally different medical aid units (ICUs), CCUs, emergency care departments (ED), medical and surgical wards, paediatric care units (PICUs), Neonatal intensive care units (NICUs), paediatric emergency care departments (PED) and paediatric inpatient\outpatient departments. Subjects: A convenience sample of 105 nurses and 105 patients was divided into 55 patients in the control group who received routine care and 50 patients in the study group who received intervention. Findings: 34.3% of studied nurses had poor knowledge pre-intervention compared with 17% post-intervention. Moreover, 33.3% of them had satisfactory knowledge pre-intervention versus 21 % post-intervention. Only 5.7% of them had excellent knowledge pre-intervention, compared with 34.4% post-intervention. Concerning complications of oxygen therapy, only 10.5% didn’t have complications in the control group versus 62.9% in the study group, 33.3% of the control group had cyanotic lips and fingernails pre intervention, versus 7.6% in the study group; 10.5% had oxygen toxicity in the control group, versus 7.6% in the study group, with a highly statistically significant difference at p 0.001 for all. Conclusion: Finally, the current results of this study concluded that there was improvement in nurses’ knowledge and practice related to oxygen therapy post intervention. Also, when the standard protocol for safe oxygen therapy was used in a positive way, it led to better health for patients and fewer problems with oxygen therapy.

Some used terms are not clear (e.g ..positive application ??,

I rephrase it, when the standard protocol for safe oxygen therapy was used in a positive way, it led to better health for patients and fewer problems with oxygen therapy.

Need to rephrase the following statement (long and confusing )

I done

Line 86, the purpose of the study is written in future tense ???

the current study evaluates the effectiveness of a standardized protocol for oxygen medical help to improve nurses' performance and patients' health outcomes.

The purpose of the study is also repeated in the method part? Why?

I delete it

Inconsistent purposes are placed in different parts of the study……. (is “Applying an intervention program “ is the purpose of the study? 

Yes

There is no need to mention questions and hypotheses…. (you either use questions or hypothesis ??)

We use hypothesis

Power analysis is highly needed in such studies to ensure that we have reached the required power of the test

Sample Size: The sample size was calculated based on statistical power of 90%, level of confidence (1-Alpha Error): 95%, Alpha 0.05, Beta 0.1. Every group determines the sample size, which is set at 30 patients. Considering 15% sample attrition, the final sample size in study group is 50 and control group 55 patients.

Is there random assignment into groups ??

No

Line 113, spelling problems “Data assortment Tools:” (it should be assessment )

Data assessment Tools

No information was provided about the reliability of the used scales (the mentioned Cronbach alpha is not accurately describe which subscale it reflects)

The Internal consistency reliability (Cronbach's α) for part II as good (.0.823), part III as good (.0.839) and part IV as good (0.845).

Need to illustrate how did the researcher sample the patients (the sampling method was only describing nurses)

Convenience sample

Inclusion and exclusion criteria for nurses and patients are not illustrated

The sample type was convenience not purposive

The distribution of nurses into control and experimental groups is not illustrated ?

Only patients who distributed on control and experimental groups

But, nurses used as one group

The researcher should explain in the method section the statistical test that has been used according to the level of measurement

Data collected from the studied sample was revised, coded and entered using Personal Computer (PC). Computerized data entry and statistical analysis were fulfilled using the Statistical Package for Social Sciences (SPSS) version 22. Data were presented using descriptive statistics in the form of frequencies, percentages and Mean SD. A correlation coefficient “Pearson correlation” is a numerical measure of some type of correlation, meaning a statistical relationship between two variables. The Kruskal-Wallis H test (sometimes also called the "one-way ANOVA on ranks") is a rank-based nonparametric test that can be used to determine if there are statistically significant differences between two or more groups of an independent variable on a continuous or ordinal dependent variable. The Wilcoxon test compares two paired groups and comes in two versions, the rank sum test, and signed rank test

Round 2

Reviewer 3 Report

Please revise the methodology section, particularly in the part of assigning participants to the control and intervention group (if you use a convenience sampling method, you should use a random assignment method when you assign participants to the control and intervention group). I suggest to add one statement in the limitation part about this 

thank you 

Author Response

Dear Reviewer :

Thank you for your valuable comments

All comments of reviewers was reply 

find attachment
